# communications
# earth & environment

# Decline in seasonal predictability potentially destabilized Classic Maya societies

Tobias Braun [1✉], Sebastian F. M. Breitenbach [2], Vanessa Skiba [1], Franziska A. Lechleitner [3], Erin E. Ray [4], Lisa M. Baldini [5], Victor J. Polyak[6], James U. L. Baldini [7], Douglas J. Kennett[8], Keith M. Prufer [4,9] & Norbert Marwan [1,10]

Classic Maya populations living in peri-urban states were highly dependent on seasonally distributed rainfall for reliable surplus crop yields. Despite intense study of the potential impact of decadal to centennial-scale climatic changes on the demise of Classic Maya sociopolitical institutions (750-950 CE), its direct importance remains debated. We provide a detailed analysis of a precisely dated speleothem record from Yok Balum cave, Belize, that reflects local hydroclimatic changes at seasonal scale over the past 1600 years. We find that the initial disintegration of Maya sociopolitical institutions and population decline occurred in the context of a pronounced decrease in the predictability of seasonal rainfall and severe drought between 700 and 800 CE. The failure of Classic Maya societies to successfully adapt to volatile seasonal rainfall dynamics likely contributed to gradual but widespread processes of sociopolitical disintegration. We propose that the complex abandonment of Classic Maya population centres was not solely driven by protracted drought but also aggravated by year-to-year decreases in rainfall predictability, potentially caused by a regional reduction in coherent Intertropical Convergence Zone-driven rainfall.

[1] Potsdam Institute for Climate Impact Research (PIK), Leibniz Association, P.O. Box 60 12 03 D-14412 Potsdam, Germany. [2] Department of Geography and Environmental Sciences, Northumbria University, Newcastle upon Tyne NE1 8ST, UK. [3] Department of Chemistry, Biochemistry and Pharmaceutical Sciences and Oeschger Centre for Climate Change Research, University of Bern, Freiestrasse 3, Bern 3012, Switzerland. [4] Department of Anthropology, University of New Mexico, Albuquerque 87131 NM, USA. [5] School of Health & Life Sciences, Teesside University, Middlesbrough TS1 3BX, UK. [6] Radiogenic Isotope Laboratory, Earth and Planetary Sciences, University of New Mexico, Albuquerque 87131 NM, USA. [7] Department of Earth Sciences, Durham University, Durham DH1 3LE, UK. [8] Department of Anthropology, University of California, Santa Barbara 93106 CA, USA. [9] Center for Stable Isotopes, University of New Mexico, Albuquerque 87131 NM, USA. [10] Institute of Geosciences, University of Potsdam, Potsdam 14476, Germany. ✉email: tobraun@pik-potsdam.de

Seasonal hydroclimate variability has defined the environmental context for tropical agricultural societies for at least seven millennia[1]. The success or failure of Late Holocene urban societies reliant on rainfed agriculture was contingent on their ability to anticipate and adapt to the seasonal distribution of rainfall from one year to the next[2]. Today, shifting rainfall seasonality due to anthropogenic climate change poses a threat to both traditional agricultural practices and food security in regions practicing rainfall dependent agriculture[3]. Studying the predictability of seasonal rainfall beyond the instrumental period using exceptionally well-dated long-term palaeoclimate archives allows us to contextualize scenarios of future dynamics due to anthropogenic climate change[4–6]. In the tropics and subtropics, rainfall seasonality has probably the strongest effect on the well-being of human society, and palaeoclimatic and archaeological studies increasingly focus on past seasonality changes using proxies from various archives, e.g., bones, molluscs, lacustrine sediments, and speleothems[4,5,7,8].

Reconstructing the impacts of palaeoclimatic variation on systems of food production, demography, and societal institutions provides examples of past climate challenges and response scenarios that are relevant to understanding future effects of global warming on these same systems[9]. Archaeologists have long had productive engagement with the palaeoenvironmental community focused on the impact of climate change on ancient societies[10]. In the tropics, food production systems are particularly vulnerable to changes in rainfall[11] which can lead to increasing conflict and destabilization of political institutions[12]. The Classic Maya are a well studied case in this regard.

In Central America, Maya urban states emerged by 900 BCE[13] following the adoption of maize as a staple grain[14] and the development of surplus agricultural production. After Maya societies in the southern lowlands (Fig. S1) underwent 1650 years of cyclical expansion and fragmentation, the period of largest demographic expansion (600-750 CE) was followed by dramatic contractions between 750-950 CE (with the strongest decline occuring between between 750 and 850 CE)[15–17] and the abandonment of large population centres led by despotically oriented dynastic lineages[18]. Maya leaders were heavily invested in wealth accumulation, kin selection, and costly ceremonial signalling that inhibited flexible and resilient responses to environmental change[19]. The inability of complex Maya societies that formed the Classic Period social and political systems to successfully respond to changing climate contributed to the geopolitical disintegration of dozens of urban centres[20] and a return to more decentralized low density agrarian villages[21]. The Terminal Classic period, defined 50 years ago, is generally described as spanning 850–1000 CE[22], though more recent literature identifies processes of societal disintegration starting a century earlier and lasting 100–150 years[23,24]. Here we describe the period of interest as the Classic Period Collapse (CPC). The 200-year CPC was a cultural process driven by increased warfare, population pressures and landscape degradation[24]. The CPC and the potential causes intrigued archaeologists and palaeoclimatologists for decades. Numerous studies present evidence for climatic disturbances as one key (though not the sole) element that led to the CPC[20,25–27]. Drier conditions between 500 and 800/850 CE coincide with increasing conflict and a general decline in the number of active political centres after 750 CE[20,28], followed by regional-scale abandonment of most of the southern Lowlands. Despite increasingly detailed palaeoclimate reconstructions and archaeological evidence, the limited historical information available from this time interval hampers our ability to deconvolve the climatic, environmental, and socio-dynamic processes that ultimately led to the CPC as we observe it. Whereas political centres in the southern Maya lowlands (Fig. S1) underwent an inexorable process of fragmentation that was not followed by the

emergence of new urban settlements, populations in the northern lowlands persisted[27,29]. The patterns of societal stress and resilience against precipitation changes are consistent with archaeological studies of climate and cultural change in different parts of the globe, including South America[30], Late Antique Turkey[31], Arabia[32], Malta[33] and Postclassic Period Mayapan (Yucatan, Mexico)[12].

During the CPC, conflicts between competing factions correlate with decadal-scale episodes of reduced rainfall[16,34–37] and enhanced hurricane activity[38]. Previous palaeoclimate studies in the Maya lowlands lacked the temporal control or sampling resolution to quantitatively assess variability in rainfall seasonality. Today, the seasonal distribution and wet season onset date are among the most critical controls for Maya farmers, who are generally small land holders producing seed and root crops for home use, animal feed and limited cash-crop surplus[39]. Modern subsistence farmers face considerable uncertainty caused by a global warming-induced decline in predictability of seasonal rainfall in recent decades. These changes are forcing adaptations in traditional agricultural practices to maximize soil moisture and hedge against increasing uncertainty in the timing of the summer monsoons[40]. Many of the staple seed and tree crops consumed by both modern and Classic Maya populations are highly vulnerable to drought conditions. During a moderate drought of 1 year without seasonal summer rainfall the number of edible plant parts available would decline by 69% including maize, beans, and squash, while in a severe multiyear period without normal summer rainfall the number of available crop foods would decline by 87%[11]. Additional vulnerability may arise from specialization of diets, impairing resilience of food systems against unpredictable year-to-year hydroclimatic conditions[23]. This suggests that instability in the seasonal distribution of rainfall, including recurring severe drought events, considerably decreases productivity in growing most staple crops consumed by pre- and post-colonial Maya populations[11], considering the severe constraints in long-term grain storage[41]. The consequences of unpredictable seasonal rainfall distributions on large geopolitical formations, with high degrees of social inequity, economic specialization, and dependence on surplus food production to feed large non-producer segments of society have not previously been addressed for the Classic Maya. Our data support the proposition that declining ability to predict the seasonal distribution of rainfall may have had profound impacts on agricultural production and, in turn, geopolitical stability of Maya population centres[2].

Here, we investigate rainfall seasonality in Central America over the past 1600 years using time series analysis on a previously published, precisely dated and sub-annually resolved speleothem record from Yok Balum Cave, southern Belize (16° 12'0 30.780"N, 89° 4'24.420"W; 336 metres above sea level)[28]. Southern Belize has some of the highest rainfall and seasonality contrasts in the Neotropics[7,42]. Due to its location near several Classic Maya centres in the southern Lowlands, Yok Balum Cave is strategically placed to explore linkages between local climate variability and cultural response (Fig. S1). Stalagmite YOK-G has a high growth rate, low age uncertainty (based on U-Th dating) and is fed by an exceptionally steady and rapid drip, allowing us to examine hydroclimate conditions during the CPC and their potential impacts on Classic Maya agricultural practises at seasonal time scales[7,43]. The speleothem stable carbon isotope ratio ($\delta^{13}C$) from YOK-G reflects local hydroclimate conditions, encompassing changes in effective infiltration above the cave and prior carbonate precipitation dynamics in the epikarst[43–45]. Additionally, stalagmite $\delta^{13}C$ is affected by soil $pCO_2$, water residence time and host rock dissolution, and $CO_2$ degassing from dripwater in the cave[46]. At our study site, all factors impacting $\delta^{13}C$ follow local hydroclimate in the same direction, thereby enhancing the link

between hydrology and $\delta^{13}C$ in the stalagmite. We focus on speleothem $\delta^{13}C$ as a proxy for local hydroclimate at seasonal scale near Maya sociopolitical centres whereas the oxygen isotope signal integrates more distal dynamics, reflecting convective activity strength, source moisture location, moisture path length, and local rainfall amount. Because YOK-G consists only of aragonite, mineral phase variability does not cause the observed isotope shifts[47].

Modern seasonal rainfall variability in the region is primarily controlled by the dynamics of the Intertropical Convergence Zone (ITCZ)[48,49]. Sub-annual rainfall distribution is distinctly seasonal, with 400–700 mm per month in the wet season (June–September) due to the northward displacement of the ITCZ during boreal summer, and 40–70 mm per month in the dry season (February–April) when the ITCZ shifts southward[42]. The spatio-temporal variability of the ITCZ is not limited to latitudinal migrations but can exhibit contractions/expansions and changes in seasonal residence times at its boundaries[28,50]. Changes in the position and latitudinal extent of the ITCZ are coupled to large-scale patterns of sea-surface temperature variability in the tropical North Atlantic[51–53] and the tropical Pacific[54]. The seasonal amplitude of rainfall in the Maya lowlands is additionally affected by tropical cyclones (TCs, July–October) and northerly winter storms (Nov–Feb)[55,56].

Several hypotheses for the climatological origins of Classic Period droughts have been proposed[57], including the latitudinal migration of the ITCZ[20,26], significant changes of tropical Northern Atlantic sea-surface temperatures[51–53,58], persistent El Niño conditions and potential interactions with TCs[55] as well as an interplay of some of these processes[52]. It has also been suggested that the ITCZ was entirely absent from the region across the Classic Maya interval, opening the door to alternative interpretations[28]. Regardless, Classic Period Maya farming systems would have been dependent on rainfall to support populations that may have grown to as large as 11 million people across the lowlands by 700 CE[59] and changes in the predictability of rainfall patterns may have negatively impacted surplus agricultural production.

The expression of such changes in climatological conditions at seasonal time scales and its repercussions on Classic Maya agriculture, however, have not yet been unravelled. Here we expand on the dynamics of local rainfall, and characterise its seasonality, year-to-year predictability of subannual rainfall distribution, and climate volatility due to extreme events. This allows unprecedented insights on potential links between rainfall seasonality and the CPC.

## Results

### Background climate and seasonal cycle.
We use advanced statistical analysis to evaluate if and how the seasonal distribution of rainfall changed over the past 1600 years in southern Belize. Dating uncertainties ($2\sigma \approx 5$ years, see Fig. S2) are propagated through the entire analysis by studying a full ensemble of COPRA age model realizations that are compatible with dating errors[60] (see methods). The YOK-G $\delta^{13}C$ record suggests a persistent drying trend initiated in 500 CE towards pronounced dry conditions between 600-800 CE. The wettest conditions occur during the Little Ice Age (LIA: 1400-1800 CE) (Fig. 1A), consistent with previous work[28]. Baldini et al. found the highest frequency of TCs during the early to mid LIA which would have likely contributed to wetter overall conditions during that period[56]. The observed multi-centennial trends are generally corroborated by both stable oxygen isotope ($\delta^{18}O$) and trace element records from the same stalagmite (see Fig. S3). Other regional records[20,26,52,61] that indicate pronounced multi-annual droughts exhibit heterogeneity

in the timing of Classic Period drought events (Fig. 1A), predominately indicating a drying trend starting after 600 CE and most severe drought events occuring between 700 and 900 CE.

Fluctuations around the long-term trends of the YOK-G $\delta^{13}C$ record (Fig. 1A, Fig. S4/S5) allow for the identification of volatile periods with strong (multi-)annual deviations from the mean climate state in contrast to periods of low variability. We use a Monte Carlo-based framework to extract indications of hydrological extreme events, i.e., exceptionally wet or dry years that reflect enhanced volatility, from the YOK-G $\delta^{13}C$ record (Fig. 1B) (see methods). A period with fewer extreme events between 550-700 CE was followed by an episode of more frequent extreme events between 700 and 900 CE, aligning with the period of most severe drought events recorded in other proxy reconstructions from the Neotropics, and tracking demographic contraction across the Maya Lowlands starting around 700 CE (Fig. 2B–D)[24]. Enhanced extreme event frequency (droughts/floods) suggests more volatile, and thus less predictable, seasonal variations on interannual time scales, beginning at a time of maximal population, increasing warfare, and high levels of social inequality[15,20]. We find two additional periods of relatively high extreme event frequency, one partly intersecting with the Medieval Climate anomaly (1100-1300 CE) and the other during the second half of the LIA (1600-1900 CE). Some indications of drought events after 1500 CE align with known historical multi-annual droughts (e.g., 1535 CE and 1765 CE[62]).

A significant positive correlation at seasonal time scales between $\delta^{13}C$ and $\delta^{18}O$ is indicative of prior calcite precipitation and/or kinetic fractionation dominating these proxies (Fig. S6).

We examine whether a seasonal cycle in the YOK-G record can be detected over the last 1600 years when dating uncertainties, irregular sampling intervals and potential aliasing are taken into account, using Lomb-Scargle periodograms and continuous Wavelet analysis (Fig. S8–12). Averaging wavelet spectral power within the 0.5–1.5 year band for each $\delta^{13}C$ age model realisation we identify age model realisations with a significant expression of seasonality (Fig. 1C). Age model realisations that indicate a significant seasonal cycle for both $\delta^{13}C$ and $\delta^{18}O$ increase markedly post-1400 CE, whereas before 1400 CE, only a muted seasonal signal is detected (further supported by Lomb-Scargle periodograms, see Fig. S12A/C), corroborating the results in ref. [28].

### Seasonal rainfall predictability.
Assuming that $\delta^{13}C$ mainly records seasonal infiltration changes and that other potential mechanisms contribute relatively little to the overall $\delta^{13}C$ changes, we can delineate the following scenarios regarding seasonal rainfall variations:

(1) Distinguishable dry and wet seasons:
Periods with distinct hydrological seasons, i.e., a distinct change in rainfall over the year, should result in lower $\delta^{13}C$ values during and shortly after the wet season, and higher $\delta^{13}C$ values during the dry season[44].

(2) Year-round wet conditions with muted dry season:
Growth rates are relatively high since dissolved inorganic carbon (DIC) is constantly supplied by incoming dripwater. Under such conditions, we expect $\delta^{13}C$ to be relatively negative, because PCP in the epikarst, and $CO_2$ degassing from drips inside the cave are minimized. At the same time, the $\delta^{13}C$ signal shows little variability at seasonal scale.

(3) Year-round dry conditions with muted wet season:
Since water and DIC supply are reduced, growth rate is expected to decline as well, which can affect sampling resolution. This in turn can further hamper predictability (see below). If droughts intensify it is possible that growth is

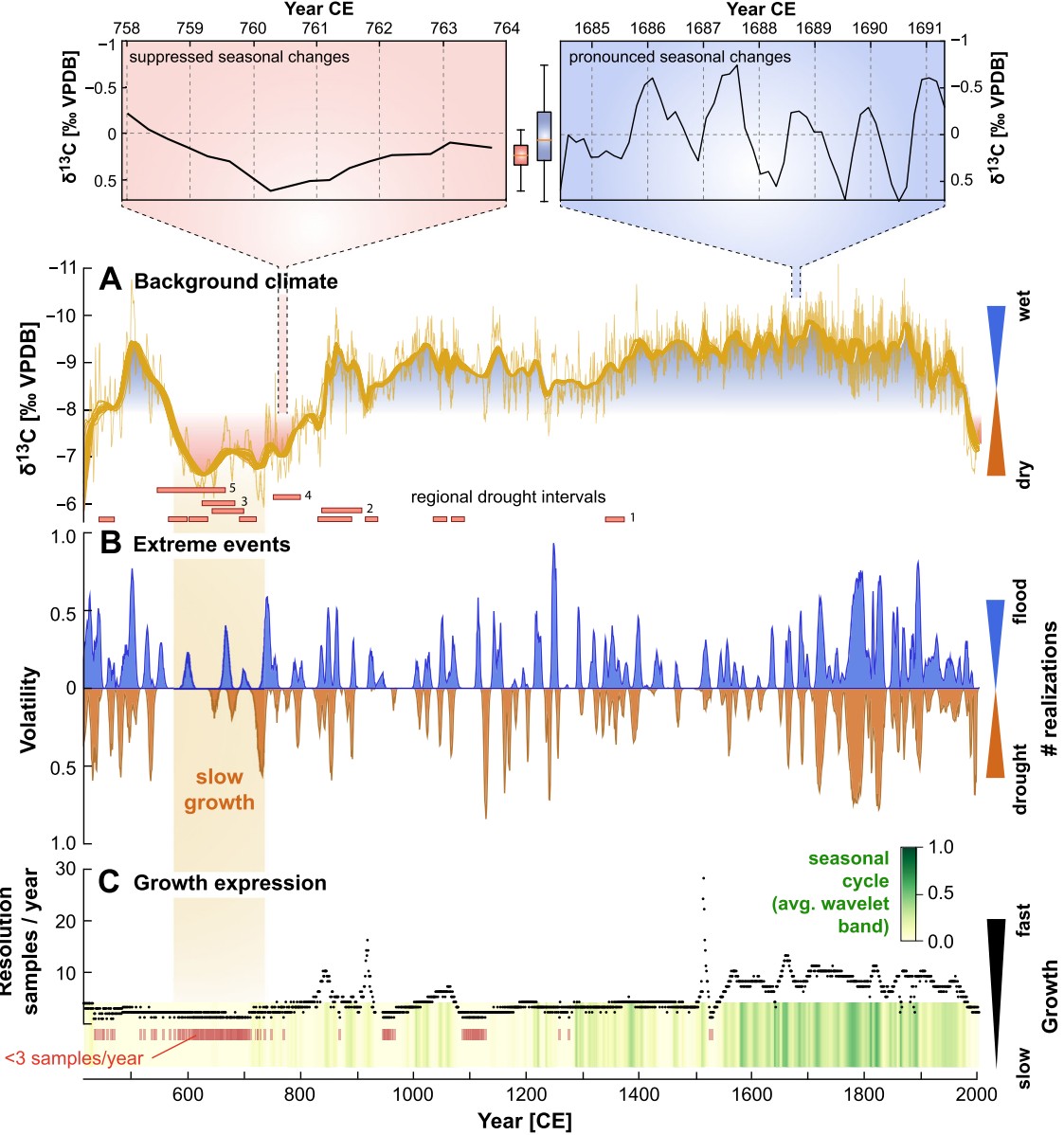

**Fig. 1 Long-term variability of YOK-G $\delta^{13}$C record and regional drought events. A** Long-term variability of the YOK-G $\delta^{13}$C record, highlighting wet (blue) and dry (red) periods in the Maya lowlands. Two insets show examples of seasonal variability. All age model realizations are detrended individually (Singular Spectrum Analysis with $w \approx 10$yr, trends for 50 realizations displayed as thick lines). The shown age model realization is the most central realization. Regional drought events from other regional records ((1) Yok-I (Yok Balum cave)[20], (2) Punta Laguna, (3) Tecoh cave, (4) Juxtlahuaca Cave, (5) Chilibrillo Cave) are indicated by red bars. Drought indications of records 2–5 are displayed according to intense dry intervals as given in ref. [52]. **B** Indications of annual extreme hydroclimate conditions. From the full ensemble of detrended $\delta^{13}$C time series, the fraction that indicates an annual drought/flood event is computed by counting how many realizations exceed the 95% quantile in each year, divided by the total number of realizations. **C** The number of samples per year represents changes in the stalagmite's growth rate. An indicator of seasonal cyclicity based on the seasonal band-average of continuous Wavelet spectra for all proxy realizations (Fig. S12) shows that episodic cyclicity is given for most segments of the record. Years with less than three samples are marked by red lines. A period of slow growth is indicated by brown shading across panels (**A**–**C**).

impeded for part of the year, which would result in micro-hiatus(es) (i.e., short growth interruptions that might last from days to months). With current sampling resolution and chronological control such micro-hiatuses will likely remain undetected.

(4) Multi-seasonal/multi-annual drought ('growth interruption'): Under severely reduced rainfall over multiple seasons, the epikarst would eventually drain and speleothem growth would cease. Just prior to such a hiatus, $\delta^{13}$C might increase due to intensifying PCP, but if the hydrological connection of the drip site reacts to a threshold value (switching from one feeding system to another), it might not record the PCP increase. Such periods would not be recorded in the $\delta^{13}$C record, but as growth interruptions.

From the perspective of Maya farmers, a regular and well-pronounced seasonal rainfall cycle allows for reliable projections of crop yield on a year-to-year basis. This entails several sets of decision making on the part of farmers. First, they need to know when to clear and prepare fields for planting. Today, this coincides with the end of the annual dry season when vegetation can be burned[27]. In much of the Maya lowlands preparation for the wet season involves clearing of vegetation, burning biomass

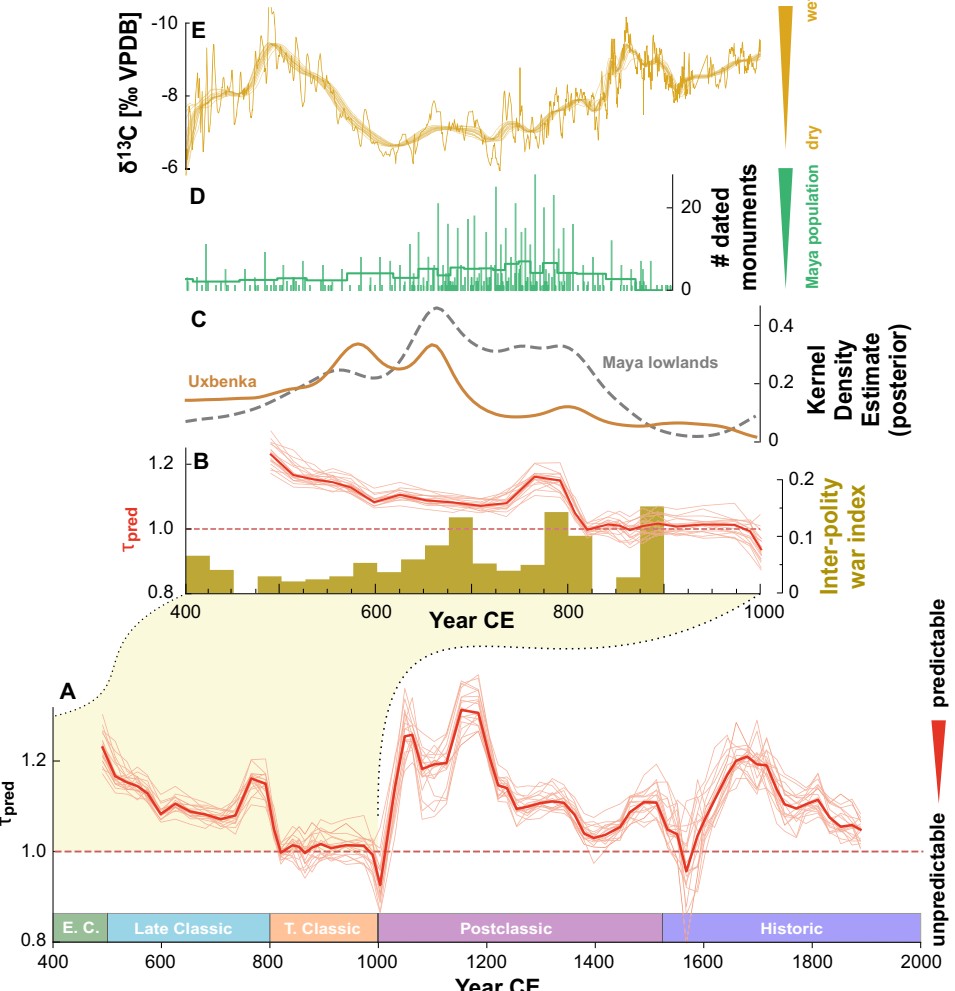

**Fig. 2 Classic Period Collapse and seasonal rainfall predictability. A** Predictability of seasonal rainfall distribution, given by relative mean predictability times $\tau_{pred}$ of 20 detrended proxy realizations for each isotope. The reference value of $\tau_{pred}^{(ref)} = 1$ (dashed line) indicates transitions between predictable states and states with a predictability that is not larger than expected from a random proxy-surrogate with the same sampling resolution. **B** A zoomed segment shows $\tau_{pred}$, indications of inter-polity war events and the evolution of Maya population size during the CPC, represented by (**C**) summed probability distributions of dated material in the Maya lowlands (grey) and the Uxbenká region (yellow) as well as (**D**) the total count of dated urban monuments. **E** Long-term drying is indicated by higher $\delta^{13}C$ values between ca. 600 and 800 CE.

for nutrients, and planting crops in anticipation of the arrival of the summer monsoons[39]. Delayed or failed arrival of the summer rains considerably increases the risk for crop failures. Classic Period farmers developed methods for continuous surplus agricultural production that was locally adapted for the diversity of environmental zones found in the lowlands[63]. This was accomplished through modifications of landscapes and the use of fire to clear land, with increasing productivity accomplished through intensification involving terracing of slopes, management of wetlands, and creation of raised beds[64,65]. However, all of these strategies relied on seasonally distributed rainfall, a dependency that increased with population size[24].

We use recurrence plots (RPs) on annually split, detrended segments of the $\delta^{13}C$ realisations to characterize the predictability of the seasonal rainfall cycle[66]. Recurrences between two annual stalagmite $\delta^{13}C$ segments signify that seasonal distribution of rainfall was similar in the respective years, indicating enhanced predictability. The recurrence-based indicator of seasonal predictability $\tau_{pred}$ employed here can be interpreted as a mean prediction time of seasonal rainfall distribution: it encodes the predictability of a year's seasonal rainfall profile based on information from the previous years' hydrological cycle. Low

average $\tau_{pred}$ values indicate a more erratic seasonal hydroclimate, requiring farmers to adapt their strategies from year to year. Sudden occurrence of a hydrological extreme event (be it drought or flood) represents one potential cause of a less predictable seasonal cycle. Most periods of low stalagmite growth also entail a less pronounced seasonal cycle in stalagmite $\delta^{13}C$ (see Fig. 1C). In Yok Balum cave, we assume that low growth rates are mainly caused by low supply of DIC. More generally, stalagmite growth can also be muted due to higher cave $pCO_2$ from reduced ventilation and higher soil respiration, entailing lower degassing rates and, in turn, reducing growth. Given the high year-round cave ventilation at our site, cave $pCO_2$ can be assumed to be low enough such that this scenario is not of high relevance here[44].

The recurrence-based approach used here is corrected for a statistical bias in seasonal predictability imposed by non-constant stalagmite growth, using the method proposed in ref. [67] (see methods). Given a stable seasonal cycle, years with seasonal conditions as described in scenarios (2) and (3) above will reduce $\tau_{pred}$ as they break the cyclical pattern and interrupt formation of diagonal lines in the RP. Scenario (2) brings along particular technical challenges that require further scrutiny. During dry conditions, seasonal $\delta^{13}C$ variations can be surpressed due to two

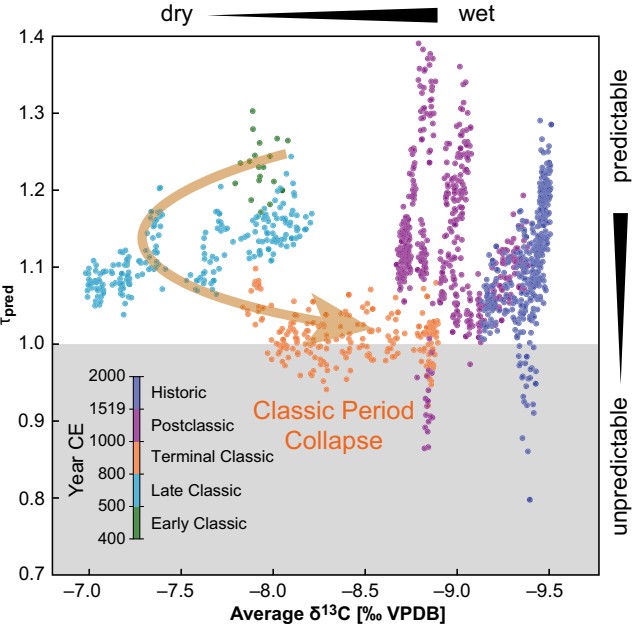

**Fig. 3 Background hydroclimate vs. seasonal rainfall predictability.** Relationship between long-term dry/wet states in background hydroclimate and seasonal predictability $\tau_{pred}$. Colour coding indicates different time periods. The reference value of $\tau_{pred}^{(ref)} = 1$ (grey shading) indicates transitions between predictable states and states with a predictability that is not larger than expected from a random proxy-surrogate with the same sampling resolution. The Terminal Classic period shows the lowest predictability prior to Postclassic times. The arrow highlights the transition from more predictable and wetter conditions to a drier and less predictable state during the late Classic, to wetter, but poorly predictable hydroclimate during the Terminal Classic.

effects: firstly, seasonal variance may be reduced due to genuinely lower seasonal amplitude, reflecting a 'true' reduction in seasonal predictability. Secondly, speleothem growth could decline which would, in turn, yield fewer samples and artificially act as a smoothing filter on the seasonal cycle. Future studies on additional regional records with higher resolution could shed light on the magnitude of this effect. Here, we estimate the potential magnitude of this effect based on a toy model as described in the supplementary material (Fig. S15) and summarized below. It should be noted, however, that when overall annual rainfall is strongly reduced, predicting arrival of seasonal rainfall is likely still impeded from a Maya farmer's perspective. Instead of reliance on timely arrival of sufficient rainfall amounts, farmers would likely adopt sowing/cropping strategies that allow mitigation of lacking water supply. If rainfall falls below mean dry season values for an extended period, crop production may be severely limited[11].

**Classic period collapse and rainfall seasonality.** The potential changes in local and regional hydroclimate discussed above may have had considerable societal repercussions during the Classic Period. We find that seasonal predictability $\tau_{pred}$ declines below the significance value of $\tau_{pred} = 1$ between 800 and 1000 CE (Fig. 2A). Such low prediction times could be generated by a random null model that simply replicates the growth rate of the stalagmite with random seasonal variation. To rule out that the observed reduction in seasonal predictability is not solely caused by reduced sampling resolution during dry periods, we employ a toy model simulation to estimate effects of sample integration/downsampling on YOK-G $\delta^{13}C$ (see Fig. S15). For both

considered downsampling scenarios, only a fraction of the observed decline in the observed predictability $\tau_{pred}$ of YOK-G $\delta^{13}C$ seasonal variations could be due to low sampling resolution. Thus, during this period irregularities at seasonal time scale are significantly large such that accounting for past seasonal rainfall does not considerably enhance the forecast for upcoming years.

The early decline in predictability suggests that rainfall became progressively harder to 'forecast' by contemporary farmers already after 500 CE, with the strongest decline in $\tau_{pred}$ after 800 CE contemporaneous to the start of the Terminal Classic. While drying as indicated by YOK-G $\delta^{13}C$ is initiated by 500 CE, the background hydroclimate is still in a relatively wet state ($\delta^{13}C < -8.5‰$) at that time (Fig. 1A). Societal processes of disintegration (increased warfare and site abandonments) were already underway between 750–830 CE in the heartland of Peten, Guatemala and Belize (Fig. 2B–C), culminating in regional depopulation by 950 CE (Fig. 2C). We argue that the observed decline in year-to-year predictability of seasonal rainfall after 500 CE could have had more immediate consequences for Maya farmers than a slow, continuous change in the hydroclimate background state.

Our results reveal a highly complex, nonlinear dependency structure between background hydroclimate and seasonal predictability (Fig. 3). This relationship is also clearly different from the relationship returned from the null-model and is thus not caused by variations in temporal sampling (Fig. S14C/D, grey dots). The most favourable conditions (i.e., highest predictability of seasonal rainfall) are identified at moderately wet conditions ($\delta^{13}C \approx -8.8‰$, Fig. 3), but no monotonous relationship (e.g., 'the wetter, the more stable') exists between average proxy values and seasonal predictability. Whenever the local or regional hydroclimate is shifting towards progressively wet conditions, either less or more seasonally predictable conditions may occur. The period from 400–550 CE is characterized by increasingly wetter conditions, followed by distinctive drying from 600-850 CE. Predictability remains high until 750 CE when it declines significantly (Fig. 2A). It reaches its minimum at 1000 CE, which is at the end of the traditional Maya Terminal Classic period, before increasing to highly predictable seasonal $\delta^{13}C$ variations between 1050–1200 CE. However, seasonal predictability deteriorates towards this minimum nearly a century after the last inscribed monuments were produced in the southern lowlands, which signalled the end of Classic Period political institutions[15]. We hypothesize that this extended period of low seasonal predictability may have inhibited a recovery of Classic period governance.

Dry intervals for which monthly precipitation does not exceed mean dry season amounts today would make it impossible for farmers to grow the vast majority of plants that make up the Maya diet[11]. Muted seasonality in a dry interval entails severe impacts on food production systems. Conversely, a loss of seasonal predictability in an overall wetter climate suggests that farming cycles would become less predictable in terms of land preparation, burning, and harvest cycles while most plants would still grow. In the 60–80 years prior to the onset of drying, the latter scenario appears conceivable.

Overall, we hypothesize that climate volatility reduced the ability of farmers to predict the onset of rainfall, leading to reduced crop yields and surpluses at the end of the Late Classic. This in turn would have impacted the ability of urban dwelling non-farmers to engage in economic activies and ensuing declines in food security may have intensified stress on political and economic institutions that ultimately led to their destabilization. Quantitative proxies for Maya population change and dated stone monuments, a proxy for governing institutions, support this interpretation (Fig. 2B/D). At Uxbenká region, an urban centre

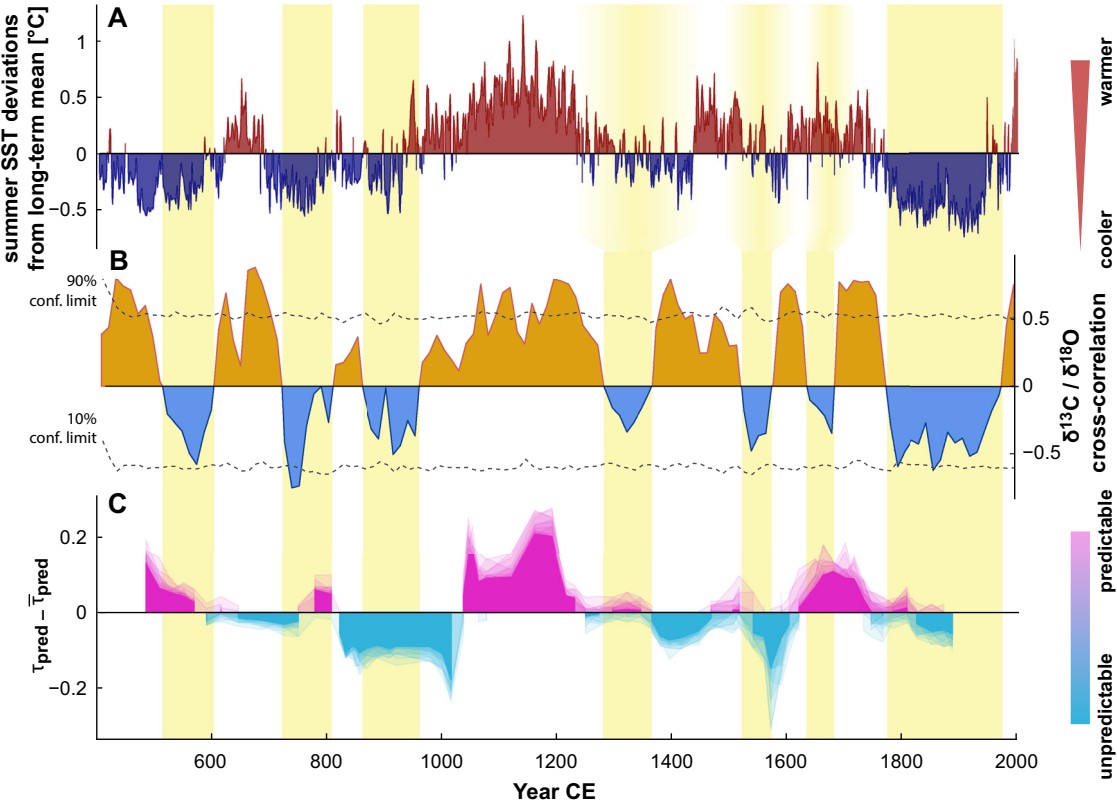

**Fig. 4 Local rainfall coherency. A** Summer SST reconstruction from the Cariaco Basin with above (below) average temperatures indicated in red (blue)[71]. **B** Local rainfall coherency index, i.e., multidecadal correlations between detrended carbon and oxygen stable isotope records which exhibit periods of significant positive and insignificant/weakly negative correlation when tested against 1000 AR(1)-realizations with a 90%-confidence level. Periods of negative correlation align well with lower SSTs (yellow shading). Lags in the alignment between summer SST and proxy correlation might be due to chronological uncertainties. **C** Deviations of seasonal predictability $\tau_{pred}$ from its long-term mean indicate predictability of local climatic conditions.

close to Yok Balum cave, this decline began around 680 CE when seasonal predictability had already deteriorated for several decades, while the final dated stone monument was erected in 780 CE, just 30 years before the site was abandoned[24]. This depopulation event aligns with demographic decline across the entire Maya lowlands starting after 700 CE and corresponds with the sharpest decline of seasonal predictability as well as severe droughts as indicated by the YOK-G record at 750 CE (Fig. 1B).

Our results suggest that a loss in seasonal predictability of rainfall may have destabilized Maya society in conjunction with severe drought events between 600 and 800 CE. Moreover, due to the continuously low predictability after 750 CE, any recovery or adaptation to new climatic states were muted. This scenario is compatible with an observed increase in the number of dated urban monuments between 700 and 800 CE as a likely response to volatile seasonal conditions. Elites, confronted with reduced surplus to finance capital projects and prestige goods likely sought to enhance their status and legitimacy as divine rulers by increasing production of carved monuments. These attested to their roles as intermediaries with important ancestors and deities deemed responsible for rainfall, social well-being, and general health[68] and were demonstrated through public ceremony, likely in lieu of more effective adaptive strategies. Reduced food security led to subordinate populations losing faith in those rulers as war-related events increased between 700 and 800 CE (Fig. 2B), reducing investment in urban and agricultural infrastructure. We suggest that the impacts of a sustained decline in seasonal predictability combined with multiple annual to decadal length droughts led to further emigration from urban areas, and an overall population decline, as well as the disintegration of >63% of urban polities with dated monuments by 835 CE[15].

**Multidecadal rainfall variability, ITCZ dynamics and the tropical North Atlantic.** What are the potential climatic drivers that modulate seasonal rainfall predictability and the frequency of droughts or floods at multidecadal time scales in the Maya region?

The multidecadal variability of Atlantic Sea surface temperatures (SST) plays a major regulatory role on seasonal rainfall variability in the Neotropics. SST changes that result in high interhemispheric thermal disparities shift the ITCZ's mean position towards the warmer hemisphere[69]. Lower SSTs across the (sub)tropical North Atlantic are less conducive to large-scale evaporation and the building of convective clouds along the ITCZ, resulting in a less coherent formation and reduced migration of the ITCZ into the northern hemisphere (NH), whereas higher SSTs have been found to increase summer precipitation in the northern Neotropics[70].

In order to assess the influence of tropical Atlantic multidecadal variability on rainfall frequency and intensity at Yok Balum Cave, we compare our seasonal rainfall predictability record with foraminifera Mg/Ca-based summer SST reconstruction from a sediment core drilled in the Cariaco basin[71] (Fig. 4A). The Cariaco basin is a very sensitive location for recording expansion/contraction or latitudinal shifts in the ITCZ as it is located in the North Atlantic beneath its northward extent[72].

We compute correlations between both YOK-G stable isotope records ($\delta^{13}C$ and $\delta^{18}O$) at seasonal (based on the most central

realization (MCR) of the age model, Fig. S6) and multi-decadal time scales (based on age model medians, Fig. S7) by extraction of suitable trends (Singular Spectrum analysis, Fig. S4). At seasonal time scales, YOK-G $\delta^{13}C$ and $\delta^{18}O$ remain significantly positively correlated throughout most of the Common Era, whereas at multidecadal time scales both periods of significantly positive, as well as periods with no relationship or significantly negative correlation exist (Fig. 4B). Periods of enhanced summer SSTs in the Cariaco basin align well with significantly positive proxy correlation in the YOK-G record and vice versa for lower SSTs, indicating a link between North Atlantic conditions and the nature of the isotope response in the stalagmite. The Medieval Climate Anomaly is marked by above average tropical Atlantic SSTs and significant proxy correlations in YOK-G. Periods of low SST and YOK-G proxy correlation occur episodically during the LIA and dominate between 1800 and 2000 CE. During the initial decline of the Classic Maya population (700–900 CE), insignificant/negative proxy correlations coincide with a period of low summer SSTs.

Previously, YOK-G $\delta^{13}C$ (reflecting rainfall amount) and YOK-G $\delta^{18}O$ (reflecting the combined influence of rainfall amount and characteristically low $\delta^{18}O$ of tropical cyclone rainfall) have been combined to isolate the tropical cyclone signal since 1550 CE[56]. Both isotope records are therefore responding to hydroclimate variability, but $\delta^{18}O$ is arguably more directly linked to North Atlantic SSTs and regional circulation patterns. Consequently, positive correlation between $\delta^{18}O$ and $\delta^{13}C$ at multidecadal time scales suggests that local hydrological conditions are in line with (pan)regional dynamics, i.e., the ITCZ and tropical Atlantic SSTs. Conversely, absence of correlation or a negative relation between both stable isotope ratios implies a control on $\delta^{18}O$ which supersedes rainfall amount, revealing the dominant influence of different (non-local) control mechanisms. Modified moisture source, trajectory, or intensity of convection along the moisture mass trajectory are likely influencing factors. Correlations between YOK-G $\delta^{18}O$ and $\delta^{13}C$ can thus be interpreted as an index for local rainfall coherency.

The ITCZ sensitively tracks changes in the interhemispheric temperature gradient. At multi-decadal time scales, changes in the ITCZ's mean position, its meridional range and its strength are all possible but remain debated for the Common Era, as argued by Asmerom et al. using the YOK-G record[28,73,74]. Most rainfall reconstructions from the Maya lowlands suggest considerable ITCZ-induced regional drying during the LIA[52]. Records from the NH do not provide convincing evidence of a large-scale latitudinal shift of the ITCZ between 600-900 CE[26,69,73,75,76]. A sound attribution of regional rainfall changes to latitudinal shifts of the ITCZ requires proxy records that cover its entire basin of influence, a spatially vast region that extends from South America to the northern Neotropics[74,77]. Compiling a sufficient number of well-dated, highly resolved records from this spatial range renders reconstructing the CPC interval challenging. However, ambiguity in CPC ITCZ reconstructions may also suggest more complexity exists above and beyond the popoular hypothesis that drying is solely driven by latitudinal shifts of the ITCZ. In fact, recent studies favour scenarios of regionally heterogeneous ITCZ responses and challenge prevailing latitudinal shift hypotheses. A compilation of regionally disparate records suggests that it is crucial to take into account additional modes of ITCZ dynamics, including expansion/contraction and region-specific weakening/intensification[74]. In particular, correlations between additional records from the Neotropics and the YOK-G $\delta^{13}C$ record studied here indicate that the ITCZ's control on the study region was considerably less dominant before 1400 CE than it is today due to ITCZ expansion/contraction[28]. Thus, whereas a cooling of North Atlantic SSTs may have triggered a

southward shift of the ITCZ during the CPC, evidence for such a shift remains ambiguous.

Based on the alignments between tropical North Atlantic SSTs, YOK-G proxy correlations, and seasonal predictability, (particularly for the CPC period (Fig. 4)), we argue that low tropical Atlantic SSTs between 700 and 900 CE resulted in declining seasonal rainfall predictability due to less coherent formation of the ITCZ over the study region. A more 'patchy' emergence of rain-producing convective activity, enhanced year-to-year variability in ITCZ residence time over the region (e.g., shortening of wet season), and stronger fluctuations in ITCZ strength would render local hydroclimate more sensitive to transient perturbations such as Caribbean TCs[56,78].

Due to extensive fractionation of uplifted water vapour, YOK-G $\delta^{18}O$ exhibits strongly depleted values during a TC[56] which could explain the observed periods of negative proxy correlations: whereas local lack of rainfall during a period of low SSTs increases both $\delta^{13}C$ and $\delta^{18}O$ values, low rainfall combined with a larger proportion of TC rainfall would result in an anticorrelation. The strength of this anticorrelation could be further accentuated by wildfires that are more likely during drier episodes and following hurricane-induced deforestation leaving dead biomass behind as fuel. The resulting reduced biomass and soil bioproductivity above the cave would increase $\delta^{13}C$, whilst TC rainfall would drive $\delta^{18}O$ strongly negative. The scenario described here would also explain an overall higher degree of volatility, reflected not only by reduced seasonal predictability $\tau_{pred}$ but also increased frequencies in annual dry/wet extremes as observed in Fig. 1B.

## Discussion

After the establishment of densely populated urban centres, Maya populations grew to a point where economic inequality, population growth, and increasing conflict would have created vulnerabilities to the effects of climate change, including changes in the seasonal distribution of precipitation. In spite of investments in water management systems, agricultural intensification, and forest management the combination of rigid political structures, economic depndencies, centuries of demographic growth, and climate change resulted in massive transformations. It has long been argued that at the peak of Classic Period developments (700 CE), the population began to push the environment towards its carrying capacity[24,26,35]. Multiple dry events have been identified as triggers for the cultural disintegration that followed the peak Classic populations as lowland Maya societies failed to produce enough food.

Our results suggest that the loss of seasonal rainfall predictability likely played a vital role in the destabilization of Classic Maya societies from the southern lowlands between 600-900 CE.

Complex societies are able to thrive despite aridity, provided that the climate is predictable, and that food production technologies have evolved to arid conditions[2,79]. We argue that the disintegration of Classic Maya society was partially catalysed by reduced predictability in climate; political institutions simply did not have measures in place to deal with irregular year-on-year changes in rainfall, sparking social unrest and inflicting societal conflicts. Based on our results, we suggest that the CPC was a period of destabilized regional climate control with reduced coherence of ITCZ-driven rainfall, giving rise to high hydroclimate volatility and reduced seasonal predictability. Today, conditions for smallholder farmers in Central America have already deteriorated considerably[80] while current projections suggest that climate change will further increase seasonal climate volatility[81]. The climate-induced disintegration of lowland Classic Maya civilization underscores the sensitivity of

human-environment systems to climatological changes, stressing the severity of drastic current global climatic changes and the urgent need to implement effective strategies that maintain food security in societies with low adaptive capacity.

## Methods

**YOK-G record**. The ca. 94 cm long stalagmite YOK-G from Yok Balum Cave in southern Belize (see ref. [44]) is composed of aragonite. The YOK-G stalagmite proxy record is based on 7151 $\delta^{13}C$ and $\delta^{18}O$ analyses, covering the last ca. 1600 years, as previously reported in ref. [28]. The chronology of the YOK-G record is based on 52 U/Th dates (ca. 0.54 per cm of growth). Although the carbon isotope record is affected by several factors (including prior carbonate precipitation (PCP) in the epikarst, $CO_2$ degassing from incoming dripwater, vegetation composition and activity, soil microbial dynamics, and temperature-induced isotope fractionation) it can be interpreted as a proxy for local effective rainfall[28,44]. The various influencing factors generally act in concert such that higher $\delta^{13}C$ values indicate drier conditions above the cave. The interpretation of the $\delta^{13}C$ profile was confirmed by uranium concentration measurements[28]. Similar to the carbon isotope ratios, the YOK-G $\delta^{18}O$ are influenced by multiple environmental processes, including moisture source and transport dynamics, rainfall amount, PCP, tropical cyclone activity, and (potentially) temperature changes[56]. However, previous studies[28,42] indicate that YOK-G $\delta^{18}O$ does reflect longer-term hydroclimate conditions above Yok Balum Cave. YOK-G $\delta^{18}O$ thus acts as proxy for (pan-)regional hydroclimate variability, with lower $\delta^{18}O$ values indicating generally wetter conditions.

**Summed probability distributions**. Summed probability distributions of radiocarbon dates (SPDs) are powerful tools for summarizing radiocarbon dates and are widely used on large datasets that cover long time periods to assess the implications of climate on cultural dynamics[82] and as population time series[83]. Traditionally, archaeologists have used counts of parameters like households, sites, or classes of artifacts as population proxies[84]. Radiocarbon dates are used to reconstruct relative changes in population sizes, generally under the assumption that (1) changes in past populations were proportional to the amount of anthropogenic carbon accumulated, and (2) the dates are distributed randomly. To many quantitative archaeologists, the patterns observed in frequency distributions of radiocarbon dates are plausible evidence of population change[83,85,86]. One weakness of this approach is the need to ascertain that event dates are indeed associated with anthropogenic activities[87]. Sample size has also been suggested to be an issue with some archaeologists indicating minimum sample size estimates[88], while others suggest that sample size thresholds should be dependent on the scale, granularity, and magnitude of specific variations of concern[85,89]. Statistical analyses suggest larger sample sizes reduce confidence envelopes[83], though many models have been criticized for both excessive noise and over-smoothing of the data to remove this noise[33,90,91]. In this study we use Bayesian inference where the prior for each observation point is effectively the kernel density estimate distribution of all the other radiocarbon dates in the dataset using the KDE_Model function in OxCal[91]. Our radiocarbon dataset employs assays drawn from the largest compendium of published radiocarbon dates from the Maya Lowlands[92] which contained 1,846 dates from 132 Maya sites. After removal of outliers, dates with large errors, and dates prior to the Classic Period our dataset consists of 1,035 assays from 80 archaeological sites, excluding Uxbenká. These dates came from excavations at political centres, settlements, caves, and cenotes representing a wide range of contexts where anthropogenic carbon accumulated over time. We supplemented this with the Uxbenká radiocarbon dataset, which consists of 338 dates from settlements and the political centre located <5 km from Yok Balum Cave[18,24,93,94].

**Historical Maya texts**. From the Early to the Terminal Classic periods Maya rulers recorded specific types of historical information, including elite political alliances, and wars, on stone monuments (stela, altar stones, and other types of dedicatory objects). Long count calendrical dates contemporaneous with the carving have been deciphered and these stone monuments can be correlated with the Gregorian calendar. Events recorded on monuments are also associated with specific long count dates. Taken together, historical records and precise dates inscribed on monuments provide an empirical foundation upon which to examine patterns of social change[95].

Following[20] we use data originally generated from the Maya Hieroglyphic Database (MHD)[96] to estimate the frequency of monument production during the Classic Period ca. 400-1000 CE. The MHD, collates monuments recording long count dates. Initial Series (IS) long count dates and calendar round dates that could be confidently correlated with the long count were only considered since they are believed to be concurrent with the original time of dedication. Focusing on the Classic Period, 882 dedicatory monuments and objects record dedicatory long count dates concurrent with time of erection from 115 sites throughout the Maya Lowlands (Fig. 2D). These monuments document >1900 events over the course of the Classic Period.

Our assessment of the frequency of warfare-related events during the Classic Period was based on keywords that relate directly to war or were commonly used by the ancient Maya describing instances of warfare, for example explicit references

to captives, warriors, destructive burning events, polity collapses; and subordinate vassalage after a defeat (terms summarized in ref. [97]).

Some mentions of war-related words that did not relate specifically to a specific warfare event that took place during the Classic Period were removed from the dataset. These included names (e.g., "He of 12 captives"), mythical events, and war-related glyphs without context. The final war-related event dataset (Fig. 2B) contained only those events involving warfare between two Maya sites or rulers, occurrences of vassalage, and other events that could be temporally grounded[20]. An index of number of warfare events to number of total events was then calculated. These datasets are plotted in Fig. 2B.

**Monte Carlo-based time series analysis**. A prerequisite to extracting seasonality from a proxy record is that it exhibits significant variability at sub- and interannual time scales. This is affected by both sampling resolution and age uncertainty; whereas the former must be high enough to assess seasonal variations (with respect to the corresponding Nyquist frequency of 2 samples/year), the latter can obscure this variability. In this work, COPRA (COnstructing Proxy Records from Age models) is used to obtain a reconstruction of $\delta^{13}C$ and $\delta^{18}O$ time series. It uses a Monte Carlo simulation scheme to generate distinct realizations of proxy values for each given value on an error-free time axis. These realizations are compatible with the limits imposed by age uncertainty and can be thought of as squeezed, stretched and translated versions of the underlying 'true' proxy time series which cannot be assessed.

COPRA transfers dating uncertainties into uncertainties in the time series magnitude[60]. The age model median time series represents a central estimate of proxy values with age uncertainty included, but much of the variability at time scales below the dating uncertainties is averaged out. We employ a simple yet effective framework to assess such time scales whereas age uncertainty is still propagated. To this extent, each of the $N$ single MC-realizations returned by the COPRA algorithm is analyzed separately by means of the statistics of interest. This technique yields $N$ values for the respective statistics which subsequently can be averaged and tested for significance with a suitable hypothesis test. In particular, we pursue this framework for identification of extreme hydroclimate conditions (Fig. 1B) and the spectral/time-frequency analysis (see Fig 1C and Fig. S8–12), all discussed in detail below. Whenever a single MC-realization is needed as a reference, we use the realization that has the highest average correlation to all other realizations and refer to it as the most central realization (MCR). Both (detrended) stable isotope MCRs are shown in Fig. S5, providing an overview of the seasonal- and decadal-scale variability.

Similar ensemble-based procedures for propagating dating uncertainties in palaeoclimate proxy time series have been carried out in other works and it is argued that this approach can reveal more information than the usual 'average age-model approach'[98]. With the focus on intra- and interannual-scale variability in this work, the individual multi-centennial trend is first extracted from each realization (see Fig. 1A) using Singular Spectrum Analysis[99] with $w = 50$ ($\approx 10$ years). At this time scale, impact by irregular sampling is small. For each of the detrended realizations, the respective statistics are computed and a null-hypothesis is defined. Significance testing is carried out to test whether the obtained values are significantly different from the null model for each MC-realization.

**Identification of annual extreme events**. As opposed to long-term dry/wet conditions, the high resolution of YOK-G $\delta^{13}C$ allows us to extract short-lived dry/wet events that are superimposed on the hydroclimate background state. The computation of block maxima/minima and the so called 'peak-over-threshold approach' are the two most popular approaches to characterize extreme events[100]. We study hydroclimate extreme conditions in YOK-G $\delta^{13}C$ based on the latter approach as it allows an adequate representation of short-lived extremes also in presence of age uncertainty. In particular, we define an upper and lower threshold based on the 95% (5%)-quantile of all detrended proxy amplitudes. Next, we check for each year how many proxy realizations exceed/fall below this threshold, indicating extreme hydroclimate conditions. As this is carried out separately for the 2000 MC-realizations of both stable isotope records, this extreme event indicator ranges between [0, 2000] and is normalized to [0, 1] by dividing by $N = 2000$ (Fig. 1B). Periods with enhanced frequency of extremely dry or wet years can be interpreted as reflecting volatile hydroclimate conditions. It has to be noted that in this analysis, no correction is applied for controlling for the variations in sampling resolution (see below).

**Time-frequency analysis**. We carry out a time-frequency analysis to examine whether a seasonal cycle can be reliably detected throughout the full stable isotope records in presence of dating uncertainties, irregular sampling and aliasing effects. Despite the very high resolution of YOK-G proxy records, age uncertainty in the range of 2–12 years (see Fig. S2) obscures spectral peaks for both $\delta^{13}C$ and $\delta^{18}O$ time series. Hence, we apply time-frequency analysis for each age model realization separately first and then tested for significance jointly to ensure that the identified cycles do not depend on a specific age model realization.

A continuous Wavelet analysis is applied to study the significance of different cycles continuously throughout the Common Era (using the PyCWT package)[101]. Since continuous Wavelet spectra (CWS) cannot deal with irregularly sampled

data, linear interpolation is applied to the proxy time series with an optimized interpolation sampling time of $\Delta t = 0.30$ years that still allows for extracting a seasonal cycle. Optimization is carried out based on Lomb-Scargle (LS) periodograms (Fig. S8 and Fig. S12/C). Significance testing is based on irregularly sampled AR(1)-surrogates (estimated from non-interpolated data) and consequently includes the variations in spectral power due to changes in the sampling resolution that are expected from an AR(1)-process. A total of 2000 CWS are computed based on each proxy realization. This yields a single 'Monte-Carlo Wavelet Spectrum' (MC-Wavelet Spectrum, Fig. S12B/D) that displays the number of proxy realizations indicating significant spectral power for a given time and frequency. Potential aliasing effects during time periods where sampling resolution in the original records falls below the Nyquist frequency (red crosses, Fig. 1C) is tested for with a simple sinusoidal model (see Fig. S11). The seasonal cycle indicator in Fig. 1C is computed from the spectral band averages (0.5–1.5 years) of all Wavelet spectra obtained from the ensemble of $\delta^{13}$C realizations. It ranges between 0 (no proxy realization indicates significant Wavelet power around the annual band) and 1 (all 2000 proxy realizations indicate significant Wavelet power around the annual band). We test how adequately the CW-analysis—despite its limitations for irregularly sampled time series—captures the relevant cycles in the record using LS periodograms[102] (Fig. S12A/C) with a Welch-overlapping sequences approach for more robust spectral estimates. We assess the significance of spectral peaks by an implementation of the REDFIT algorithm in Python 3.7[103].

**Recurrence analysis.** We perform recurrence analyses to characterize the varying degree of predictability of the sub-annual rainfall distribution. Recurrence analysis is based on the computation of a (binary) recurrence matrix $\mathbf{R}$ that indicates the recurrence between two states at times $i$ and $j$ by a value of 1 while 0 represents no recurrence between those values. Recurrence plots are a powerful tool in the analysis of nonlinear time series that, for example, provide quantitative measures to detect regime shifts or identify couplings[66,104].

Two time series values are regarded as 'recurrent' if their distance $D_{ij}$ falls below a certain threshold $\epsilon$. In this work, $\epsilon$ was chosen such that $\mathbf{R}$ is always filled with 10% of recurrences[105]. Diagonal lines in an RP signify periods in which two segments of the time series show similar variations, i.e., one segment is predictable from the other. We choose the mean diagonal line length of an RP as a quantifier of seasonal predictability. It can be interpreted as a mean predictability time, quantifying for how many subsequent years (on average) a sub-annual rainfall distribution could be predicted from other years.

The major obstacle of comparing the sub-annual distribution of different years is imposed by irregular sampling. Standard procedures to compute the distance matrix $\mathbf{D}$ (e.g., Euclidean distance) cannot be utilized. Moreover, the number of samples per year is correlated to the growth rate of the stalagmite and thus represents additional meaningful information on the hydrological conditions. We employ the edit-distance method to compute distances between sub-annual segments to account for the irregular sampling[106]. The fundamental idea of this method is to estimate the distance between two segments by means of a transformation cost, i.e., a sum of basic operations that need to be applied to the one segment to transform it into the other. Time series values can only be shifted (both in time and amplitude), deleted or added whereas a certain cost needs to be specified for each of these operations. The costs associated with shifting must both include a cost related to the magnitudes and the time instances. In ref. [107], a modification was introduced to account for the fact that the costs of shifting in time should saturate beyond a certain time scale $\tau_e$. The (m)edit distance between two segments $S_a$ and $S_b$ resulting from these added costs should be minimized over all possible combinations of operations and is given by

$$D(S_a, S_b) = \min_C \left\{ \sum_{\alpha, \beta \in C} \underbrace{f_{\Lambda_0(t(\alpha), t(\beta); \tau_e)}}_{\text{shifting}} + \underbrace{\Lambda_k \| L_a(\alpha) - L_b(\beta) \|}_{\text{amplitude change}} + \underbrace{\Lambda_S(|I| + |J| - 2|C|)}_{\text{adding and deleting}} \right\}$$

with the $\alpha$-th/$\beta$-th amplitudes $L_a(\alpha), L_b(\beta)$ of the segments $S_a, S_b$ and the cardinalities $|\cdot|$ of the sets $I, J$ and $C$. $D(S_a, S_b)$ is a metric distance. The cost parameters $\Lambda_0, \Lambda_k$ and $\Lambda_S$ need to be fixed prior to cost optimization. The sigmoid function $f_{\Lambda_0}(t(\alpha), t(\beta); \tau_e)$ introduced in ref. [107] controls the costs of time shifting. We follow the empirical estimation procedures proposed in ref. [106] to fix all parameters.

The (m)edit-distance method is applied to annual time series segments ($w_{\text{(m)edit}} = 1$ year) on 200 years-sliding windows (25% overlap). Due to irregular sampling, the length of sub-annual segments (i.e., values per year) changes (see Fig. 1C). In ref. [67], a systematic effect of such variations on the resulting (m)edit-distance was identified and examined. In context of a recurrence analysis, this effect needs to be corrected which is carried out following the proposed correction scheme: we generate 20 sampling rate-constrained surrogates (SRC-surrogates) for each proxy realization and the time axis returned from the COPRA age model. Each SRC-surrogate reproduces the numbers of samples per year while sampling intervals and proxy amplitude differences are drawn randomly, yielding a new set of synthetic time axis and time series. This way, the systematic sampling rate bias and its relation to variance in the amplitudes are conserved. A recurrence plot is computed for each SRC-surrogate following the procedure outlined above

(examples: Fig. S13). For each RP, we compute a mean diagonal line length $T_{\text{pred}}$ that represents the mean prediction time of the given time window (in years). Windows are centred at the respective time instances. We account for the effect of nonstationary sampling rate by defining the relative prediction time:

$$\tau_{\text{pred}} = \frac{T_{\text{pred}}}{T_{\text{pred}}^{(\text{surr})}}$$

with $T_{\text{pred}}^{(\text{surr})}$ as the 95%-quantile of the mean prediction time distribution obtained from the SRC-surrogate distribution. Whenever $\tau_{\text{pred}} > 1$, sub-annual proxy variations can be interpreted as predictable for longer time periods than expected solely from variations of the sampling resolution. Age uncertainty is partly included by repeating the analysis for 20 randomly selected, distinct proxy realizations due to computational constraints. Additional effects that result from stalagmite growth rate variations but are untreated by this method are conceivable, such as sampling integration or a wet season bias. We estimate the expected effect that these two factors could exert on seasonal predictability using MC-realizations of a sinusoidal toy model (see Fig. S15).

## Data availability
The YOK-G record has been published in ref. [28]. We provide the studied YOK-G records and, specifically, the age model ensembles used here in the following Zenodo repository: https://doi.org/10.5281/zenodo.7104976

## Code availability
A collection of Python functions and scripts is provided via Zenodo. These can be used to replicate the manuscript's main figures: https://doi.org/10.5281/zenodo.7104976

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

## Acknowledgements

Both Yok Balum cave and Uxbenká are located on the community owned traditional lands of the Mopan Maya people of Santa Cruz, Toledo, Belize located with the Maya homeland of southern Belize. Data derived from climate proxies at Yok Balum Cave and from archaeological contexts at Uxbenká were collected with permission from of the successive elected leaders of the community, and the Uchben'kaj Kin Ajaw Association to KMP. The authors thank the community for our long collaboration on this project. Permits to remove samples and conduct analysis were obtained from from the Belize Institute of Archaeology to KMP. This research was supported by the Deutsche Forschungsgemeinschaft in the context of the DFG project MA4759/11-1 'Nonlinear empirical mode analysis of complex systems: Development of general approach and application in climate'. VS is supported by DFG grant FO 809/6-1. FL acknowledges support from the Swiss National Science Foundation (SNSF) grant PZ00P2_186135. Funding for the archaeological research at Uxbenká, the initial production of the Yok G and Yok I climate records, and ongoing research related to these records came from the US National Science Foundation (BCS BCS-0620445, Prufer, HSD 0827305 Prufer, Kennett, and Polyak), the Alphawood Foundation (2010–2016 Prufer). We would also like to thank the three reviewers for their valuable feedback.

## Author contributions

T.B. designed the study, performed the analysis, led the writing, and prepared the manuscript. K.M.P. initiated the joint work. S.F.M.B., F.A.L., L.B., J.U.L.B., D.J.K. and K.M.P. coordinated the synthesis. K.M.P., D.J.K. and E.E.R. led the archaeological interpretations. V.S., K.M.P. and N.M. contributed data and/or analysis. T.B. and N.M. developed the methods. V.J.P. carried out the measurements. S.F.M.B., T.B., N.M. and V.S. contributed to preparation of figures. S.F.M.B. and N.M. supervised the study. All authors provided critical feedback and helped shape the research, analysis and manuscript.

## Funding

## Competing interests

The authors declare no competing interests.
