## [Peer Review File · Communications Earth & Environment]

1st Nov 22

Dear Mr Braun,

First of all, we apologize for the delay in processing your manuscript. Your manuscript titled "Decline in seasonal predictability potentially destabilized Classic Maya societies" has now been seen by 3 reviewers, and I include their comments at the end of this message. They find your work of interest, but some important points are raised. We are interested in the possibility of publishing your study in Communications Earth & Environment, but would like to consider your responses to these concerns and assess a revised manuscript before we make a final decision on publication.

We therefore invite you to revise and resubmit your manuscript, along with a point-by-point response that takes into account the points raised. Please highlight all changes in the manuscript text file.

Specifically, we would like to see the authors focus on the following editorial thresholds:

- 1) Provide more discussion on the factors affecting seasonal variation in the speleothem record
- 2) Address the weaknesses of the SPD method
- 3) Provide an updated bibliographic references (as suggested by Reviewer 3) and revise the mislabeled figures.

Please use the following link to submit your revised manuscript, point-by-point response to the referees' comments (which should be in a separate document to any cover letter) and the completed checklist:

[link redacted]

We hope to receive your revised paper within six weeks; please let us know if you aren't able to submit it within this time so that we can discuss how best to proceed. If we don't hear from you, and the revision process takes significantly longer, we may close your file. In this event, we will still be happy to reconsider your paper at a later date, as long as nothing similar has been accepted for publication at Communications Earth & Environment or published elsewhere in the meantime.

We understand that due to the current global situation, the time required for revision may be longer than usual. We would appreciate it if you could keep us informed about an estimated timescale for resubmission, to facilitate our planning. Of course, if you are unable to

estimate, we are happy to accommodate necessary extensions nevertheless.

Please do not hesitate to contact me if you have any questions or would like to discuss these revisions further. We look forward to seeing the revised manuscript and thank you for the opportunity to review your work.

Best regards,

Michael Storozum, PhD
Editorial Board Member
Communications Earth & Environment
orcid.org/0000-0002-3948-6074

Clare Davis, PhD
Senior Editor
Communications Earth & Environment

EDITORIAL POLICIES AND FORMATTING

Editorial Policy: [Policy requirements](https://www.nature.com/documents/nr-editorial-policy-checklist.pdf) (Download the link to your computer as a PDF.)

Furthermore, please align your manuscript with our format requirements, which are summarized on the following checklist:

[Communications Earth & Environment formatting checklist](https://www.nature.com/documents/commsj-phys-style-formatting-checklist-article.pdf)

and also in our style and formatting guide [Communications Earth & Environment formatting guide](https://www.nature.com/documents/commsj-phys-style-formatting-guide-accept.pdf) .

***** DATA:** Communications Earth & Environment endorses the principles of the Enabling FAIR data project (<http://www.copdess.org/enabling-fair-data-project/>). We ask authors to make the data that support their conclusions available in permanent, publically accessible data repositories. (Please contact the editor if you are unable to make your data available).

All Communications Earth & Environment manuscripts must include a section titled "Data Availability" at the end of the Methods section or main text (if no Methods). More information on this policy, is available at <http://www.nature.com/authors/policies/data/data-availability-statements-data-citations.pdf>.

If a community resource is unavailable, data can be submitted to generalist repositories such as [figshare](https://figshare.com/) or [Dryad Digital Repository](http://datadryad.org/). Please provide a unique identifier for the data (for example a DOI or a permanent URL) in the data availability statement, if possible. If the repository does not provide identifiers, we encourage authors to supply the search terms that will return the data. For data that have been obtained from publically available sources, please provide a URL and the specific data product name in the data availability statement. Data with a DOI should be further cited in the methods reference section.

REVIEWER COMMENTS:

Reviewer #2 (Remarks to the Author):

The manuscript declared that the reduced predictability of seasonal rainfall led to the collapse of agricultural production during the Terminal Classic Maya period, which in turn destabilized Classic Maya societies. The literature on climate change and its connections with classic Maya culture has been extensively studied. In this paper, instead of directly linking drought to reduced food surplus, the authors examine the climatic volatility and societal repercussions. What's more, the analysis methods are advanced, including Kernel Density Estimate, recurrence analysis and so on. The authors quantitatively calculated the predictability of rainfall season, which show clear evidence for the reduced rainfall season predictability coinciding with evidence for violence and population decline. This article introduces several innovative and convincing ideas. However, I still have some main concerns.

1. The main idea of this paper is that the reduced predictability of precipitation seasons led to Terminal Classic collapse (TCC). The article highlights the importance of predictability, but the

variation of precipitation seasonality is not clearly described. For example, why does $\delta^{13}\text{C}$ shows an insignificant seasonal cycle during TCC, what's the corresponding seasonal hydroclimate characteristic, is the absence of the rainy season? And how it affects τ predictability? There should be more discussion of the factors influencing the seasonal variation of the stalagmite proxy.

2. Second, the authors highlight that precipitation seasonal predictability decreases during the TCC, possibly influenced by the missing precipitation seasonal cycle and the increased frequency of extreme events. However, the conclusion may be biased due to the irregular sampling resolution, since the magnitude and the seasonal signal are significantly influenced by the sampling resolution. The recurrence analysis seems only to consider methods for analyzing irregularly sampled data. The authors should further consider how to eliminate the indistinct seasonal cycles and small amplitudes potentially caused by the low sampling resolution. On the other hand, $\delta^{13}\text{C}$ records reach 1994CE, why is the predictability τ only around 1900CE?

3. The article declares that "a pronounced decline in the predictability of seasonal rainfall starting prior to the onset of previously documented protracted drought conditions in the neotropical Maya lowlands". Why compare the record of predictability reconstructed in this paper with drought records (700-900CE) reconstructed by other work? According to YOK-G $\delta^{13}\text{C}$ records, the driest period was 500-700CE, while the precipitation predictability decreased significantly at 800-1000CE. So, the decrease in predictability was after the drought period rather than before.

There are some other suggestions:

"As this is carried out separately for the 2000 MC-realizations of both stable isotope records, this extreme event indicator ranges between [0; 2000] and is normalized to [0; 1] by dividing by $N = 2000$ ". There should be more introduction about the method to identification of annual extreme events.

"In this work, ...R is always filled with 15 % of recurrences". Is there any reference to determine the threshold?

Reviewer #3 (Remarks to the Author):

This study applies statistical analyses to a previously published, extremely well dated and high resolution speleothem record from Belize in order to investigate links between changes in Mayan society and seasonal to inter-annual hydroclimate variability, primarily associated with the ITCZ. Several previous papers have highlighted many of the main points of this work and as such it is a bit derivative. However, it is very well constructed and interesting and does provide some novel information (e.g., most predictably seasonal rainfall coincident with moderately wet conditions) and some interesting new hypotheses (e.g., impacts of hurricanes and fire on speleothem $\delta^{13}\text{C}$). The methods are clearly explained, straightforward, and appropriate. It's a really nice paper that, in my opinion, is ready to publish as is. The only question is whether the editor decides this is sufficiently novel to merit publication by CE&E. A few comments:

Why not include line numbers?

Swap the order of the first two sentences in the abstract.

“The 200-year Terminal Classic Collapse (TCC) was a cultural process driven by increased warfare, population pressures and landscape degradation” – Haug et al. (2003) tie elevated abundances of Ti and Fe in Cariaco basin sediments to climatic change at this time. Why not include a reference to climate variability as a driver of the TCC?

Reviewer #4 (Remarks to the Author):

In the paper “Decline in seasonal predictability potentially destabilized Classic Maya societies”, Braun and colleagues present an excellent discussion on the impacts of seasonal predictability of precipitation for the Classic Maya. The seasonal data is based on statistical re-analysis of previously-published speleothem stable isotope data (from a paper including some of the same contributors). The author team contains experts in all relevant fields, and this is reflected in the high quality of the manuscript. Furthermore, the topic of the paper is consistent with the aims of *Communications: Earth & Environment* and it provides multiple important contributions to several fields, which I really appreciate. I recommend this valuable paper is published within *Communications: Earth & Environment* following some minor revisions.

The first of the important contributions is your novel methodology used for the speleothem proxy re-analysis. I am not an expert on the statistical methods employed, but they were explained clearly in the methods/ supplementary file and there were no clear problems with the methodology. These methods bring greater value to records and should be used more widely by palaeoclimatologists!

Secondly, the focus on seasonality as opposed to just precipitation is fantastic. The Kwiecien paper you cite is great (thanks!) and should perhaps be discussed a bit more. Further, other examples of seasonality from proxy data could be worth mentioning – stable isotope analysis of different materials from Lake Nar comes to mind (Dean et al., 2018) but there are more proxies and discussions of seasonality by Amy Prendergast (et al., 2018) etc.

Thirdly, emphasis on the human experience/perception of the changes (via agriculture and “predictability”). This is great and you could cite some of the key historical/archaeological theory papers that claim the human element needs to be emphasised. I suspect there are regional/Maya specific examples of this, but I am thinking of e.g., Haldon et al., 2018

Fourthly, the combination of multiple factors undermining resilience thus leading to significant societal change – the “perfect storm”. This has been a recurring theme in the last couple of years – it might be worthwhile referencing some works from elsewhere in the globe that found this complexity. For example, in Late Antique Turkey (Jacobson et al., 2022) and Arabia (Fleitmann et al., 2022), as well as in Malta during the 4.2ka event (Groucutt et al., 2022).

Figures:

The figures are great and easy to interpret. They all contain a large amount of information but display this concisely. I especially appreciate the zoomed in sections at the top of Fig 1

and on the Fig 2 timescale that visualised points made in the text. Fig 3 is also a great way to visualise the changes and explained very well in the caption. However, I do have two small recommendations. Firstly, the regional drought events in Fig 1 do not need full citations in the figure as the papers (#13 and #38) are cited in the caption. Secondly, Fig 2 has the y-axes labelled in reverse when compared to Figs 1 and 4. I would switch this around for consistency.

There are a few other minor concerns which should be easy to address:

- Reading the paper, it seemed as though you were underselling its value! The important contributions outlined above should be given more emphasis and should also be better contextualised by referencing papers with similar findings. I have given some recommendations for papers to cite, but there are likely more appropriate ones from closer to your study region.
- Speleothem YOK-G is aragonitic as opposed to calcitic. This is not a problem, but I think the potential influence of this on the record should be mentioned.
- Summed radiocarbon – there should be more transparency here with the weaknesses of this method. There is a clear bias here from which sites/periods have been studied thus far, as well as impacts from uncertainties (see Carleton and Groucutt, 2021 and other papers by Groucutt). I don't think this means the method should be avoided altogether, but its weaknesses should be more clearly portrayed.

Thank you for this amazing piece of work, I am very grateful to have been able to review it. –
Matthew J Jacobson, University of Glasgow, UK

References

Carleton, W. C., & Groucutt, H. S. (2021). Sum things are not what they seem: Problems with point-wise interpretations and quantitative analyses of proxies based on aggregated radiocarbon dates. *The Holocene*, 31(4), 630–643.

<https://doi.org/10.1177/0959683620981700>

Dean, J.R.; Jones, M.D.; Leng, M.J.; Metcalfe, S.E.; Sloane, H.J.; Eastwood, W.J.; Roberts, C.N. Seasonality of Holocene Hydroclimate in the Eastern Mediterranean Reconstructed Using the Oxygen Isotope Composition of Carbonates and Diatoms from Lake Nar, Central Turkey. *Holocene* 2018, 28, 267–276, <https://doi.org/10.1177/0959683617721326>.

Fleitmann, D.; Haldon, J.; Bradley, R.S.; Burns, S.J.; Cheng, H.; Edwards, R.L.; Raible, C.C.; Jacobson, M.J.; Matter, A. Droughts and Societal Change: The Environmental Context for the Emergence of Islam in Late Antique Arabia. *Science* 2022, 376, 1317–1321, <https://doi.org/10.1126/science.abg4044>.

Groucutt, H.S.; Carleton, W.C.; Fenech, K.; Gauci, R.; Grima, R.; Scerri, E.M.L.; Stewart, M.; Vella, N.C. The 4.2 Ka Event and the End of the Maltese “Temple Period.” *Front. Earth Sci.* 2022, 9, 1–23, <https://doi.org/10.3389/feart.2021.771683>.

Haldon, J.; Mordechai, L.; Newfield, T.P.; Chase, A.F.; Izdebski, A.; Guzowski, P.; Labuhn, I.; Roberts, N. History Meets Palaeoscience: Consilience and Collaboration in Studying Past

Societal Responses to Environmental Change. *Proc. Natl. Acad. Sci. U. S. A.* 2018, 115, 3210, <https://doi.org/10.1073/PNAS.1716912115>.

Jacobson, M.J.; Pickett, J.; Gascoigne, A.L.; Fleitmann, D.; Elton, H. Settlement, Environment, and Climate Change in SW Anatolia: Dynamics of Regional Variation and the End of Antiquity. *PLoS One* 2022, 17, e0270295, <https://doi.org/10.1371/journal.pone.0270295>.

Prendergast, A.L.; Pryor, A.J.E.; Reade, H.; Stevens, R.E. Seasonal Records of Palaeoenvironmental Change and Resource Use from Archaeological Assemblages. *J. Archaeol. Sci. Reports* 2018, 21, 1191–1197, <https://doi.org/10.1016/j.jasrep.2018.08.006>.

Potsdam Institute for Climate Impact Research P.O. Box 60 12 03 D-14412 Potsdam

Anonymous reviewers
Communications Earth & Environment

Submission of revised manuscript COMMSENV-22-0621

Dear reviewers,

Thank you for your friendly and helpful reviews from 1st November 2022. We appreciate you taking the time to read our manuscript as well as providing detailed feedback. We have carefully reviewed the comments and have revised the manuscript accordingly. Our responses are given in a point-by-point manner below. Changes to the manuscript and the supplementary material are highlighted in red/blue. Line numbers refer to the versions with highlighted changes.

Best regards

Tobias Braun

**POTSDAM INSTITUTE FOR
CLIMATE IMPACT RESEARCH**

P.O. Box 60 12 03
D-14412 Potsdam | Germany
T: +49 331 288 2500
F: +49 331 288 2600
www.pik-potsdam.de

Member of the Leibniz Association

Scientific Directors:
Prof. Dr. Ottmar Edenhofer
Prof. Dr. Johan Rockström

Association Registry Number:
Local Court Potsdam VR 1038 P

Bank account:
Mittelbrandenburgische
Sparkasse Potsdam (MBS)
IBAN: DE69 1605 0000 3502 2355 29
BIC: WELADED1PMB

12th December 2022

Tobias Braun
+49 331 288 20744
tobraun@pik-potsdam.de

Response to Reviewer 2:

Thank you for your critical and helpful review of our paper. We address each of your points below (line numbers refer to the versions with highlighted changes).

Reviewer:

The main idea of this paper is that the reduced predictability of precipitation seasons led to Terminal Classic collapse (TCC). The article highlights the importance of predictability, but the variation of precipitation seasonality is not clearly described. For example, why does $\delta^{13}C$ shows an insignificant seasonal cycle during TCC, what's the corresponding seasonal hydroclimate characteristic, is the absence of the rainy season? And how it affects the predictability? There should be more discussion of the factors influencing the seasonal variation of the stalagmite proxy.

Authors:

We agree with the reviewer that this discussion was addressed too briefly in the previous version of the manuscript. We adjusted our discussion to explain in more detail how changes in seasonal rainfall would be expressed in the proxy record, and how this would affect the proposed predictability index τ_{pred} . To this extent, we discuss several possible scenarios of how different seasonal rainfall conditions would be reflected in YOK-G $\delta^{13}C$. The adjusted discussion can be found in the main text of the revised manuscript in lines 367 (left) – 363 (right) of the manuscript and p.14 of the supplementary material (referring to the versions with highlighted changes).

Reviewer:

Second, the authors highlight that precipitation seasonal predictability decreases during the TCC, possibly influenced by the missing precipitation seasonal cycle and the increased frequency of extreme events. However, the conclusion may be biased due to the irregular sampling resolution, since the magnitude and the seasonal signal are significantly influenced by the sampling resolution. The recurrence analysis seems only to consider methods for analyzing irregularly sampled data. The authors should further consider how to eliminate the indistinct seasonal cycles and small amplitudes potentially caused by the low sampling resolution. On the other hand, $\delta^{13}C$ records reach 1994CE, why is the predictability τ only around 1900CE?

Authors:

The reviewer addresses one of the main technical challenges of this work, that is, accounting for irregular sampling and related biases. The edit distance method, along with our method to 'normalize' T_{pred} using surrogate time series with the same sampling resolution as the original record, enable us to study seasonal predictability over the full record despite variations in the sampling resolution. The problem of 'indistinct seasonal cycles and small amplitudes potentially caused by the low sampling resolution' as raised by

the reviewer, however, justifies further scrutiny. In principle, seasonal $\delta^{13}\text{C}$ variations in some dry years could solely be muted due to sampling integration. While such effects can only be fully ruled out with additional (ultra-)highly-resolved records, we designed a toy model experiment to evaluate this issue. The proposed toy model assumes constant predictability but variations in the sampling rate and is able to simulate potential biases on τ_{pred} . Using this toy model, we establish an estimator for the potential impact of sampling effects (sampling resolution) on seasonal variations in YOK-G $\delta^{13}\text{C}$. We briefly discuss the results from these simulations in l.423-457 (left) and 422-436 (right) of the main manuscript. A few lines have also been added to the corresponding methods paragraph. A comprehensive description of the model and obtained results is given in the suppl. material p.15-16 (Fig. S15). We are confident that this analysis provides additional confidence to our key finding and thank the reviewer for this important suggestion.

Finally, the fact that the time axis only reaches 1900 CE is due to the windowing inherent to our statistical methodology (and holds also true for the beginning of the record). The windows are centered at the respective time points and thus also include information beyond 1900 CE. The sequence of overlapping windows was chosen in this way to avoid biases towards the edges of the record. This information has been added in the methods.

Reviewer:

The article declares that “a pronounced decline in the predictability of seasonal rainfall starting prior to the onset of previously documented protracted drought conditions in the neotropical Maya lowlands”. Why compare the record of predictability reconstructed in this paper with drought records (700-900CE) reconstructed by other work? According to YOK-G $\delta^{13}\text{C}$ records, the driest period was 500-700CE, while the precipitation predictability decreased significantly at 800-1000CE. So, the decrease in predictability was after the drought period rather than before.

Authors:

The reviewer is correct in that there is a significant decline in predictability after 800 CE, i.e., well after the onset of driest conditions as recorded in YOK-G $\delta^{13}\text{C}$. However, we observe that predictability had already declined, beginning around 500 CE, well prior to any major droughts, i.e., 60-80 years prior to the onset of the driest interval (as indicated by highest $\delta^{13}\text{C}$ values) (between ca. 570 CE and 800 CE). We argue that such a decline would potentially yield very immediate consequences for Maya farmers and their cropping/sowing strategies. The hydroclimate at this stage is still in a rather wet state. We have, however, also carefully refined all statements on the conjunction of drought events and seasonal predictability. We discuss all of these findings now more explicitly in l.482 (left) – 461 (right).

Reviewer:

“As this is carried out separately for the 2000 MC-realizations of both stable isotope records, this extreme event indicator ranges between [0; 2000] and is normalized to [0; 1] by dividing by $N = 2000$ ”. There should be more introduction about the method to identification of annual extreme events.

Authors:

We agree that the description of how extreme events are identified is rather brief. We have added a few additional sentences that add on the fundamental idea, method and interpretation in the methods (l.868-885 (right)).

Reviewer:

“In this work, ...R is always filled with 15 % of recurrences”. Is there any reference to determine the threshold?

Authors:

A reference has been added. In the cited work, choosing the threshold based on quantiles of the distance distribution has been studied extensively. Also, we noticed that this was not reported correctly. The recurrence plots are actually filled with 10% of recurrences. This has been corrected. In any case, it has been ensured that the result remain robust for reasonable variations of this parameter.

Additional note:

Beyond the reviewers' comments, we have compiled a repository that contains the full dataset and scripts that are required to fully replicate main figures of this study. The repository is documented and freely available through Zenodo as described in the updated data and code availability statements in the manuscript.

Response to Reviewer 3:

The authors thank the reviewer for the positive and helpful feedback. Each of your suggestions is addressed below (line numbers refer to the versions with highlighted changes).

Reviewer:

Why not include line numbers?

Authors:

We did not include line numbers because they are unfortunately not provided in the template. We have now included them.

Reviewer:

Swap the order of the first two sentences in the abstract.

Authors:

We agree with the reviewer that this will enhance readability and have swapped them.

Reviewer:

“The 200-year Terminal Classic Collapse (TCC) was a cultural process driven by increased warfare, population pressures and landscape degradation” – Haug et al. (2003) tie elevated abundances of Ti and Fe in Cariaco basin sediments to climatic change at this time. Why not include a reference to climate variability as a driver of the TCC?

Authors:

We have added some additional sentences that refer to works that follow similar lines of research, attributing the TCC to a climate origin (l.127-141 (left), l.116-141 (right)). Here, we cite references that are already contained in the bibliography as the latter is already very large. One of the strengths of this paper is that we move beyond simple wet/dry dichotomies to look at seasonal variation. We note that if wet season rainfall declined until it was equal to dry season amounts it would make it difficult for Maya farmers to produce surplus foods and many crops would not be grown or consumed.

Additional note:

Beyond the reviewers' comments, we have compiled a repository that contains the full dataset and scripts that are required to fully replicate main figures of this study. The repository is documented and freely available through Zenodo as described in the updated data and code availability statements in the manuscript.

Response to Reviewer 4:

We would like to thank the reviewer for his enthusiastic, comprehensive and helpful feedback! We are glad to hear that the reviewer finds our study to be a valuable contribution. Please find our responses below (line numbers refer to the versions with highlighted changes).

Reviewer:

Secondly, the focus on seasonality as opposed to just precipitation is fantastic. The Kwiecien paper you cite is great (thanks!) and should perhaps be discussed a bit more. Further, other examples of seasonality from proxy data could be worth mentioning – stable isotope analysis of different materials from Lake Nar comes to mind (Dean et al., 2018) but there are more proxies and discussions of seasonality by Amy Prendergast (et al., 2018) etc.

Authors:

We would like to thank you for your enthusiastic comment regarding seasonality. Thank you also for the additional references. We added another sentence with references in lines 119-126 (left) of the main text to highlight the relevance of seasonality for agricultural societies.

Reviewer:

Thirdly, emphasis on the human experience/perception of the changes (via agriculture and “predictability”). This is great and you could cite some of the key historical/archaeological theory papers that claim the human element needs to be emphasised. I suspect there are regional/Maya specific examples of this, but I am thinking of e.g., Haldon et al., 2018

Authors:

We thank the reviewer for the suggestions. We have now supplemented the cited references by additional relevant historical/archaeological papers in the introduction (l.127-141 (left), 148-155(left/right)). We hope that this better contextualizes our findings and puts further emphasis on the Maya farmer perspective on seasonal rainfall changes.

Reviewer:

Fourthly, the combination of multiple factors undermining resilience thus leading to significant societal change – the “perfect storm”. This has been a recurring theme in the last couple of years – it might be worthwhile referencing some works from elsewhere in the globe that found this complexity. For example, in Late Antique Turkey (Jacobson et al., 2022) and Arabia (Fleitmann et al., 2022), as well as in Malta during the 4.2ka event (Groucutt et al., 2022).

Authors:

Thank you for this comment. We have addressed this in a synthesis added to our discussion of Maya climate and culture (see our response above).

Reviewer:

The figures are great and easy to interpret. They all contain a large amount of information but display this concisely. I especially appreciate the zoomed in sections at the top of Fig 1 and on the Fig 2 timescale that visualised points made in the text. Fig 3 is also a great way to visualise the changes and explained very well in the caption. However, I do have two small recommendations. Firstly, the regional drought events in Fig 1 do not need full citations in the figure as the papers (#13 and #38) are cited in the caption. Secondly, Fig 2 has the y-axes labelled in reverse when compared to Figs 1 and 4. I would switch this around for consistency.

Authors:

Thank you for your positive comments on the figures. We have erased the full citations in the figure panel in Fig. 1. We understand that the labeling of Fig. 2 is 'upside down' compared to all other figures and thus inconsistent. However, turning the labels around appears illogical and would likely make it less clear as all top panels are zoom-outs. We would like to keep them in the current order and hope that the reviewer agrees with our view.

Reviewer:

Reading the paper, it seemed as though you were underselling its value! The important contributions outlined above should be given more emphasis and should also be better contextualised by referencing papers with similar findings. I have given some recommendations for papers to cite, but there are likely more appropriate ones from closer to your study region.

Authors:

We highly appreciate the reviewer's perspective on the value of our paper. We believe that we have improved on the contextualization of our study by the extended discussion of regional archaeological findings in the literature, as suggested by the reviewer. We hope that, with the provided elaborations and additional discussions, we get the value of our study across to the readers.

Reviewer:

Speleothem YOK-G is aragonitic as opposed to calcitic. This is not a problem, but I think the potential influence of this on the record should be mentioned.

Authors:

We did not discuss this in the paper because we have only a single mineral

phase (aragonite) in YOK-G. The mineral phase (aragonite vs. calcite) of the stalagmite is not a problem for the stable isotope record since no changes between both mineral phases are found in the sample. The latter would indeed affect carbon and oxygen isotope ratios because of the difference in fractionation factors between aragonite-water and calcite-water [Fohlmeister, J. *et al.* Carbon and oxygen isotope fractionation in the water-calcite-aragonite system. *Geochimica et Cosmochimica Acta* 235, 127–139 (2018)]. A detailed overview on the correction of isotope data from a stalagmite from Mayapan, northern Yucatan, is available in the supplementary information of [Kennett, Douglas J., et al. "Drought-Induced Civil Conflict Among the Ancient Maya." *Nature communications* 13.1 (2022): 1-10.]. We added a short note in the discussion for clarification (lines 209-212 (left)).

Reviewer:

Summed radiocarbon – there should be more transparency here with the weaknesses of this method. There is a clear bias here from which sites/periods have been studied thus far, as well as impacts from uncertainties (see Carleton and Groucutt, 2021 and other papers by Groucutt). I don't think this means the method should be avoided altogether, but its weaknesses should be more clearly portrayed.

Authors:

We thank the reviewer for this comment, which mirrors concerns expressed in modern demographic studies. We added a discussion to the manuscript (l.836 (left) - 829 (right)) that evaluates both the strengths and weaknesses in methods employing summed radiocarbon dates generally and specially the strengths in our dataset.

Additional note:

Beyond the reviewers' comments, we have compiled a repository that contains the full dataset and scripts that are required to fully replicate main figures of this study. The repository is documented and freely available through Zenodo as described in the updated data and code availability statements in the manuscript.

31st Jan 23

Dear Mr Braun,

Your manuscript titled "Decline in seasonal predictability potentially destabilized Classic Maya societies" has now been seen by our reviewers, whose comments appear below. In light of their advice I am delighted to say that we are happy, in principle, to publish a suitably revised version in Communications Earth & Environment under the open access CC BY license (Creative Commons Attribution v4.0 International License).

We therefore invite you to revise your paper one last time to comply with our format requirements and to maximise the accessibility and therefore the impact of your work.

EDITORIAL REQUESTS:

*****Please take care to match our formatting and policy requirements. We will check revised manuscript and return manuscripts that do not comply. Such requests will lead to delays. *****

SUBMISSION INFORMATION:

OPEN ACCESS:

Communications Earth & Environment is a fully open access journal. Articles are made freely accessible on publication under a [CC BY license](http://creativecommons.org/licenses/by/4.0) (Creative Commons Attribution 4.0 International License). This license allows maximum dissemination and re-use of open access materials and is preferred by many research funding bodies.

For further information about article processing charges, open access funding, and advice and support from Nature Research, please visit <https://www.nature.com/commsenv/article-processing-charges>

At acceptance, you will be provided with instructions for completing this CC BY license on behalf of all authors. This grants us the necessary permissions to publish your paper. Additionally, you will be

asked to declare that all required third party permissions have been obtained, and to provide billing information in order to pay the article-processing charge (APC).

[link redacted]

Best regards,

Michael Storozum, PhD
Editorial Board Member
Communications Earth & Environment

Clare Davis, PhD
Senior Editor
Communications Earth & Environment

www.nature.com/commsenv/
@CommsEarth

REVIEWERS' COMMENTS:

Reviewer #2 (Remarks to the Author):

Thank you for your careful revisions. I think all my concerns are addressed in the revised version.

Reviewer #3 (Remarks to the Author):

This already solid work has been substantially improved by these changes. It's quite an impressive piece of work and the presentation is splendid!

Reviewer #4 (Remarks to the Author):

Thank you for your revisions and responses. I am happy with the corrections and edits made to this great paper! Where my recommendations were not accepted, the authors have adequately justified their choices. I recommend no further revisions and will look forward to reading the published article